# Exploring the Potential Mechanism of Prothioconazole Resistance in *Fusarium graminearum* in China

**DOI:** 10.3390/jof9101001

**Published:** 2023-10-10

**Authors:** Feng Zhou, Aohui Han, Yan Jiao, Yifan Cao, Longhe Wang, Haiyan Hu, Runqiang Liu, Chengwei Li

**Affiliations:** 1Henan Engineering Research Center of Green Pesticide Creation and Pesticide Residue Monitoring by Intelligent Sensor, Henan Institute of Science and Technology, Xinxiang 453003, China; zfhist@163.com (F.Z.); aohuihan123@163.com (A.H.); j1917007935@163.com (Y.J.); 15936146642@163.com (Y.C.); 18736126649@163.com (L.W.); 2Henan Engineering Research Center of Crop Genome Editing/Henan International Joint Laboratory of Plant Genetic Improvement and Soil Remediation, Henan Institute of Science and Technology, Xinxiang 453003, China; haiyanhuhhy@126.com; 3School of Food Science and Engineering, Henan University of Technology, Zhengzhou 450001, China; 4Henan Engineering Research Center of Biological Pesticide & Fertilizer Development and Synergistic Application, Henan Institute of Science and Technology, Xinxiang 453003, China

**Keywords:** *Fusarium graminearum*, prothioconazole, fungicide resistance, resistance mechanism, cross-resistance

## Abstract

The Fusarium head blight (FHB) caused by *Fusarium graminearum* is one of the most important diseases threatening wheat production in China. However, the triazole sterol 14α-demethylation inhibitor (DMI), prothioconazole, is known to exhibit high activity against *F. graminearum*. The current study indicated that three highly resistant laboratory mutants exhibited significantly (*p* < 0.05) altered growth and sporulation, although contrary to expectation, only one of the mutants exhibited reduced growth and sporulation, while the other two exhibited significant (*p* < 0.05) increases. Despite this, pathogenicity tests revealed that all of the mutants exhibited significantly (*p* < 0.05) reduced pathogenicity, indicating a substantial cost to fitness. Sequence analysis of the prothioconazole target protein, CYP51, of which *F. graminearum* has three homologues (FgCYP51A, FgCYP51B, and FgCYP51C), identified three mutations in the FgCYP51B sequence with a high likelihood of being associated with the observed resistance, as well as another three mutations in the FgCYP51B sequence, and two in the FgCYP51A sequence that are worthy of further investigation. Two of the prothioconazole-resistant mutants were also found to have several amino acid substitutions in their FgCYP51C sequences, and it was interesting to note that these two mutants exhibited significantly (*p* < 0.05) reduced pathogenicity compared to the other mutant. Expression analysis revealed that prothioconazole treatment (0.1 μg/mL) resulted in altered expression of all the *FgCYP51* target genes, and that expression was also altered in the prothioconazole-resistant mutants compared to their wild-type parental isolates. Meanwhile, no evidence was found of any cross-resistance between prothioconazole and other commonly used fungicides, including carbendazim, pyraclostrobin, and fluazinam, as well as the triazole tebuconazole and the imidazole DMI prochloraz. Taken together, these results not only provide new insight into potential resistance mechanism in *F. graminearum*, and the biological characteristics associated with them, but also convincing evidence that prothioconazole can offer effective control of FHB.

## 1. Introduction

Wheat is one of the most important global food crops, providing about 21% of dietary calories and 20% of human protein requirements due to its high nutritive value [1]. A recent report by the Food and Agriculture Organization of the United Nations (FAO) (https://www.fao.org/faostat/-en/#data/QCL, accessed on 9 November 2022) indicates that world wheat production in 2021 was in the region of 770.88 million tons, and it has been noted that the stability of wheat yields is critical to ensure world food security and economic development [2]. In China, wheat constitutes one of the three traditional staple foods and plays a strategic role in the steady development of the national economy. The main wheat-growing region, Huang-Huai-Hai, spans five provinces, including Henan, Shandong, Hebei, Tianjin, and Beijing, and accounts for approximately 57.54% of the area under wheat cultivation and 63.10% of the total output [3]. However, the Fusarium head blight (FHB), caused primarily by *Fusarium graminearum*, poses a constant threat to the production of wheat and other cereals, both in China and throughout the world [4,5]. Indeed, FHB is considered an economically important disease of global significance that causes extensive damage to wheat crops, particularly in humid and semihumid regions of the world, especially Canada, Australia, Asia, and South America [4,6]. Infection with FHB not only greatly reduces crop yields, but also the kernel quality as a result of contamination with mycotoxins such as deoxynivalenol that render the grains unsuitable for either human food [7,8] or livestock feed [9]. The planting of FHB-resistant varieties is generally considered the preferred strategy for protecting vulnerable crops on account of the low economic cost and minimal environmental risk [5]. Chinese wheat cultivars from the Jiangsu and Hubei Provinces, in the Middle and Lower Yangtze River Valleys, had the highest levels of FHB resistance, whereas those from other provinces, including Shandong, Henan, and Hebei in the Huang-Huai-Hai region (where almost half of the nation’s wheat is produced), tended to have much lower levels of resistance [10]. In the absence of highly resistant cultivars, these growers have relied heavily on fungicide sprays to provide protection against FHB over the last few decades [4,6], with carbendazim, pydiflumetofen, phenamacril, azoxystrobin, and prothioconazole being registered for the control of FHB in China (http://www.chinapesticide.org.cn/, accessed on 3 October 2023). Such fungicides are therefore of significance in safeguarding the nation’s wheat production.

Prothioconazole is a broad-spectrum systemic fungicide belonging to the triazole family of sterol 14α-demethylation inhibitors (DMIs) that was developed by the Bayer Crop Protection Company in 2004 [11], and the first registered in China for control of FHB in 2019 (http://w.icama.cn/zwb/dataCenter?hash=reg-info, accessed on 3 October 2023). Providing excellent systemic activity against *Fusarium* and other plant pathogens [12,13], prothioconazole has been widely used in agriculture to protect cereal, bean, and root crops [14,15,16,17]. The DMI mode of action relies on inhibition of the 14α demethylase enzyme, CYP51, which is essential for the biosynthesis of ergosterol [18,19]. More specifically, CYP51 is known to catalyze demethylation at C14 of lanosterol (LAN) or 24-methylene dihydrolanosterol (DHL) during the production of ergosterol, which is an essential component of the fungal cell membrane [20,21,22]. However, as a consequence of inappropriate and overuse of DMI fungicides, there have been numerous reports of resistance emerging to this important group of fungicides, with resistance being documented in 40 plant pathogens, and confirmed field resistance being reported in 28 species [23]. The emergence of resistance to prothioconazole has been well characterized over the past 20 years, and it has been noted that a reduction in performance typically becomes evident after only a few years of intensive use [4]. Meanwhile, research regarding the underlying causes of resistance has identified three primary mechanisms of resistance to DMI fungicides: (i) changes to the amino acid sequence of the CYP51 protein that result in reduced affinity for azoles [24,25]; (ii) overexpression of the *CYP51* genes as a result of changes to the upstream promoter region [26,27]; and (iii) over-expression of efflux pumps, including ATP-binding cassette (ABC) transporters and those belonging to the major facilitator superfamily (MSF) of transporters, which can reduce sensitivity by expelling DMI fungicides from the fungal cells [28]. The current study was initiated in order to provide a better understanding of the mechanisms leading to prothioconazole resistance in *F. graminearum* via the use of laboratory mutants. The study investigated both the biological characteristics of the mutants as well as possible resistance mechanisms using sequence analysis of the *FgCYP51* genes and quantitative expression experiments. In addition, the study evaluated the potential for cross-resistance between prothioconazole and other commonly used fungicides.

## 2. Materials and Methods

### 2.1. Fungicides, Isolates, and Media

Most of the technical-grade fungicides (Appendix A) used in the study, including prothioconazole, tebuconazole, prochloraz, pyraclostrobin, and fluazinam, were dissolved in acetone (analytically pure, Tianjin Fuyu Fine Chemical Co., Ltd., Tianjin, China), with the exception of carbendazim, which was dissolved in 0.1 mol/L of hydrochloric acid (HCl, analytically pure, Tianjin Fuyu Fine Chemical Co., Ltd.). The stock solutions were stored at 4 °C for no longer than two weeks before serial dilutions were freshly prepared for each experiment. Mycelial growth assays were performed to confirm that none of the solvents had any effect on the growth of *F. graminearum* at the range of concentrations tested.

The three wild-type *F. graminearum* isolates identified using the molecular biology of their internally transcribed spacer (ITS) used in the current study included 3-a, 4-a, and SZ-1-3, which were originally collected from infected wheat ears exhibiting typical symptoms of FHB in the fields of Henan Province, China in 2019 (Table 1). All of the wild-type isolates were highly sensitive to prothioconazole, with EC_50_ values of 0.579, 0.311, and 0.251 μg/mL, respectively. After repeated exposure of these isolates to prothioconazole under laboratory conditions following the protocol of a previous study [29]), three genetically stable, highly resistant mutants (3-aR, 4-aR, and SZ-1-3R) were recovered, which had EC_50_ values of 10.10, 12.34, and 21.24 μg/mL, respectively (Table 1). Both the wild-type isolates and the resistant mutants were routinely maintained on potato dextrose agar (PDA: 20 g/L glucose, 200 g/L potato, and 20 g/L agar) at 24 °C.

### 2.2. Mycelial Growth, Sporulation, and Pathogenicity of Prothioconazole-Resistant Mutants of F. graminearum

The biological characteristics of the three resistant *F. graminearum* mutants (3-aR, 4-aR, and SZ-1-3R) were compared to those of their wild-type parental isolates (3-a, 4-a, and SZ-1-3), by assessing mycelial growth on PDA, sporulation in mung bean broth, spore germination in sterilized water, and pathogenicity on wheat coleoptiles using the methods detailed in a previous study [29]. Each isolate was represented by at least eight individual plates in the growth assay, six flasks in the sporulation and spore germination assays, and ten wheat coleoptiles in the pathogenicity tests, with each experiment being performed twice.

### 2.3. Cloning and Sequencing of Three FgCYP51 Genes, Including FgCYP51A, FgCYP51B, and FgCYP51C

Mycelium samples were prepared following the method of a previous study [29], and total genomic DNA was extracted using the Omega bio-tek Fungal DNA Kit (Omega bio-tek lnc., Guangzhou, China) according to the instructions of the manufacturer. The resulting DNA was used as a template for amplification of the full-length *FgCYP51* genes (*FgCYP51A*, FGSG_04092; *FgCYP51B*, FGSG_01000, and *FgCYP51C*, FGSG_11024), with primer sets designed to the three different homologues using primer premier software (ver.6.0., PREMIER Biosoft, Canada): FgCYP51A-F/FgCYP51A-R, FgCYP51B-F/FgCYP51B-R, and FgCYP51C-F/FgCYP51C-R, respectively (Appendix A). The PCR was performed using 50 μL reaction mixtures containing 25 μL of 2×ES Taq Master Mix, 1.5 μL of template DNA, 2 μL of each primer, and 21.5 μL of ddH_2_O (CoWin Biosciences, Cambridge, MA, USA). It was processed using a 96-well thermal cycler (Applied biosystems, Thermo Fisher Scientific, Waltham, MA, USA) with the following program: initial denaturation at 95 °C for 2 min; followed by 35 cycles of melting at 95 °C for 30 s; annealing at 57 °C for 90 s in the case of *FgCYP51A* (57 °C for *FgCYP51B* and 57.5 °C for *FgCYP51C*); and extension at 72 °C for 80 s in the case of *FgCYP51A* (85 s for *FgCYP51B* and 95 s for *FgCYP51C*); with a final extension at 72 °C for 10 min. The resulting PCR products were then purified and cloned into the pMD19-T vector before being sequenced commercially (Wuhan Genecreate Biotechnology Co. Ltd., Wuhan, China). The *FgCYP51A*, *FgCYP51B*, and *FgCYP51C* sequence data obtained from each isolate or mutant were analyzed with DNAMAN software (Ver.8.0. Lynnon Biosolf, CA, USA) and compared using multiple sequence alignments.

### 2.4. Relative Expression of Three FgCYP51 Genes in Prothioconazole-Resistant Mutants of F. graminearum

Total RNA was extracted from fresh mycelial samples that had been cultured in both the absence and presence of prothioconazole (0.1 μg/mL) using a fungal RNA kit (Omega bio-tek, Basel, Switzerland) according to the protocol of the manufacturer. First-strand cDNA was synthesized using the PrimeScript RT reagent kit (TaKaRa, Kusatsu, Japan) and used as a template for the qPCR amplification of partial sequences (approximately 200 bp in length) of the *FgCYP51A*, *FgCYP51B*, and *FgCYP51C* genes using the RT-FgCYP51A-F/RT-FgCYP51A-R, RT-FgCYP51B-F/RT-FgCYP51B-R, and RT-FgCYP51C-F/RT-FgCYP51C-R primer sets, respectively (Appendix A). The qPCR itself was performed using the QuantStudio 6 Flex PCR detection system (ThermoFisher, Waltham, MA, USA), with reaction mixtures containing SYBR Green I fluorescent dye, and the following program: initial denaturation at 95 °C for 10 s; followed by 40 cycles of 95 °C for 5 s, and 60 °C for 32 s; and dissociation at 95 °C for 15 s, 60 °C for 60 s, and 95 °C for 15 s. The relative expression of each gene was then determined following the protocol of a previous study using actin as the reference gene. Each isolate or mutant was represented by three biological replicates, and the entire experiment was performed twice [29].

### 2.5. Cross-Resistance between Prothioconazole and Other Commonly Used Fungicides

The potential for cross-resistance between prothioconazole and other commonly used fungicides was assessed using the protocol described in a previous study [29]), with the following fungicide concentrations: 0, 0.0005, 0.0015, 0.0045, 0.0135, 0.0405, 0.1215, 0.3645, 1.0935, and 3.2805 μg/mL for tebuconazole, prochloraz, carbendazim, pyraclostrobin, and fluazinam, as well as prothioconazole when assessing the wild-type parental isolates; and 0, 1.5625, 3.125, 6.25, 12.5, 25, 50, and 100 μg/mL for prothioconazole with the prothioconazole-resistant mutants. Each treatment was represented by three separate plates, and the entire experiment was performed twice.

### 2.6. Statistical Analysis

The data collected in the mycelial growth, sporulation, spore germination, and pathogenicity experiments, as well as the results from gene expression analysis, were subjected to analysis of variance (ANOVA) using SPSS software ver. 17.0 (SPSS Inc., Chicago, IL, USA), with significant differences between treatments being determined using Fisher’s least-significant difference test (*p* ≤ 0.05).

## 3. Results

### 3.1. Mycelial Growth, Sporulation, and Pathogenicity of Three Prothioconazole-Resistant Mutants of F. graminearum

All of the mutants exhibited significant (*p* < 0.05) changes in the biological characteristics assessed, although there was great variation in the degree and type of change. For example, two of the mutants (4-aR, and SZ-1-3R) had an increased growth rate on PDA, while the other (3-aR) displayed reduced growth (Figure 1). A similar pattern was observed for sporulation, with 4-aR and SZ-1-3R producing significantly (*p* < 0.05) more spores than their parental isolates, double the number in the case of 4-aR, while 3-aR produced significantly (*p* < 0.05) less, with a 70% reduction (Figure 2A). However, it was found that the germination rate of the spores produced by 3-aR was actually significantly (*p* < 0.05) increased, whilst that of 4-aR was significantly reduced (Figure 2B). Similar to 3-aR, the spores of SZ-1-3R also exhibited an increased rate of germination. Meanwhile, all of the prothioconazole-resistant mutants were found to exhibit a significant reduction in pathogenicity in comparison to their wild-type parental isolates (Figure 3A), although the degree of change varied, with 3-aR exhibiting the most reduced lesions (≈70% smaller), and SZ-1-3R the least reduced (≈40% smaller) (Figure 3B). Taken together these results indicate that although prothioconazole resistance was associated with a certain cost to fitness, particularly with regard to pathogenicity, there was also a high degree of variation between the different mutants, with some mutants even exhibiting an increase in fitness with regard to growth and sporulation, a phenomenon that might be a reflection of the underlying molecular biology associated with the specific resistance mechanism of each mutant. 

### 3.2. Sequence Analysis of Three FgCYP51 Genes in Prothioconazole-Resistant Mutants of F. graminearum

The *FgCYP51* sequences of the prothioconazole-resistant mutants were found to contain numerous point mutations when compared to those of their parental isolates (Table 2), which incidentally were identical to the sequences of the *F. graminearum* wild-type strain (PH-1) detailed in the GenBank database. Although several of the changes were found to be silent mutations that did not affect the predicted amino acid sequences, many others resulted in amino acid substitutions that might be associated with the observed prothioconazole resistance. For example, the 3-aR mutant was found to have substitutions in all three of its FgCYP51 sequences, including L16F and S35P in FgCYP51A; P74T, F77L, and G405S in FgCYP51B; and E164K, I191T, S256A, M273L, and V424E in FgCYP51C. Meanwhile, the other two mutants were found to have fewer mutations, with 4-aR having only one substitution (Y230F) in its FgCYP51B sequence and one (V307A) in its FgCYP51C, and the SZ-1-3R mutant had just two mutations (Y37N and Q326R) in its FgCYP51B sequence. Despite these results, this study failed to identify any conserved mutations that were present in all three mutants that might be a strong indication of their contribution to a potential resistance mechanism. Indeed, none of the mutants were found to share even a single amino acid substitution or loci that might indicate a critical residue, although all three did have mutations in their FgCYB51B homologue, which has previously been associated with DMI resistance in *F. graminearum* [25,30].

### 3.3. Relative Expression of Three FgCYP51 Genes in Prothioconazole-Resistant Mutants of F. graminearum

The relative expression of the *FgCYP51* genes exhibited a large degree of variation, not only between the prothioconazole-resistant mutants and their sensitive parental isolates, but also in the absence or presence of the fungicide, and even between the three genes themselves (Figure 4). Consequently, it was difficult to identify any clear patterns of expression that might be associated with a potential prothioconazole resistance mechanism. However, some notable observations could still be made. For example, the expression of *FgCYP51A* was the only gene that produced a consistent pattern of expression in all of the mutant and parental isolates tested, with the prothioconazole treatment resulting in a significant increase in expression (Figure 4A). Indeed, in some cases, the fungicide caused as much as a 5–10-fold increase in expression in the wild-type isolates (4-a and SZ-1-3). However, the effect was generally less pronounced in the prothioconazole-resistant mutants (4-aR and SZ-1-3R), although a similarly large increase in expression was observed in 3-aR, even though its parental isolate (3-a) had much reduced expression compared to the other wild-type isolates. With regard to *FgCYP51B*, there was almost no consistency in the pattern of expression, with the prothioconazole treatment resulting in significantly (*p* < 0.05) reduced expression in the wild-type isolates 3-a and 4-a, and dramatically increased expression (≈10-fold) in SZ-1-3 (Figure 4B). A similar lack of consistency was also observed in the prothioconazole-resistant mutants, although it was interesting to note that one mutant (3-aR) exhibited a significantly (*p* < 0.05) reduced basal level of expression compared to its parental isolate (3-a), while the other two (4-aR and SZ-1-3R) had a significantly (*p* < 0.05) higher level of basal expression in the absence of prothioconazole. The most notable aspect regarding the expression of the *FgCYP51C* gene was that one of the parental isolates (SZ-1-3) exhibited no observable expression either in the absence or presence of the fungicide (Figure 4C). However, interestingly, its mutant progeny (SZ-1-3R) had a basal level of expression comparable to the other mutant and wild-type isolates but had the highest level of expression in the presence of prothioconazole. Although these results confirm that prothioconazole treatment results in altered expression of the *FgCYP51* target genes, and that expression was altered in the prothioconazole mutants, without further study, it was difficult to form any strong conclusions regarding how this might affect the prothioconazole response of the wild-type isolates, or how it might relate to any potential resistance mechanisms in the resistant mutants. 

### 3.4. Cross-Resistance between Prothioconazole and Other Commonly Used Fungicides

Although the current study found that prothioconazole exhibited high activity against wild-type isolates of *F. graminearum* (3-a, 4-a, and SZ-1-3), with EC_50_ values of 0.58, 0.31, and 0.25 μg/mL, respectively, this effect was greatly reduced in the resistant mutants (3-aR, 4-aR, and SZ-1-3R), which had values of 10.09, 12.34, and 21.24 μg/mL (Table 1). However, despite this dramatic change in sensitivity, no evidence of cross-resistance was found between prothioconazole and any of the other fungicides tested, including tebuconazole, which is also a triazole fungicide, and prochloraz, which is an imidazole DMI (Table 1).

## 4. Discussion

FHB is a devastating disease that affects wheat crops worldwide and is a constant threat to wheat production in China, leading to greatly reduced yields, as well as poor grain quality that threatens the health of humans and livestock as a result of mycotoxin contamination [10,31]. Although FHB can be partially managed via good agronomic practice, at present, the application of fungicides is still the primary method of control in the absence of stable FHB-resistant varieties [10,29]. However, inappropriate and overuse of commonly used fungicides for an extended period of time has already led to the emergence of field resistance to many of the most effective fungicides, including the benzimidazole fungicide carbendazim and the triazole fungicide tebuconazole, in Chinese populations of *F. graminearum* [25,29,32]. 

Prothioconazole is a novel broad-spectrum DMI fungicide that was recently developed by the Bayer Crop Protection Company, which has high activity against many fungal diseases that affect cereal and legume crops [11,14,15,16,17]. However, to date, prothioconazole has yet to be specifically registered for the control of FHB in China, perhaps on account of fears that reliance on a single compound to tackle this disease might result in the rapid emergence of resistance, although it has been registered for use in combination with carbendazim or tebuconazole (http://www.chinapesticide.org.cn/zwb/dataCenter, accessed on 3 October 2023). Despite such fears, a recent survey of 255 *F. pseudograminearum* isolates collected from the Henan province of China found no evidence of resistance, and that all the field isolates remained highly sensitive to prothioconazole [33]. The study also found that prothioconazole-resistant mutants of *F. pseudograminearum* produced under laboratory conditions exhibited reduced growth and sporulation, as well as reduced pathogenicity on wheat [33]. In contrast, the results of the current study found that only one mutant of *F. graminearum* (3-aR) exhibited a similar reduction in growth and sporulation, whilst the other two (4aR and SZ-1-3) exhibited significantly (*p* < 0.05) increased growth and sporulation rates compared to their parental isolates (Figure 1 and Figure 2A). Despite this, the current study found that all of the mutants still exhibited significantly (*p* < 0.05) reduced pathogenicity in wheat seedling (Figure 3), indicating that, similar to *F. pseudograminearum* [33], prothioconazole resistance also leads to a significant fitness cost in *F. graminearum*.

As a typical triazole fungicide, prothioconazole is known to inhibit sterol biosynthesis in plant pathogenic fungi [11], specifically the 14α-demethylase enzyme, CYP51. This mode of action, which is directed at a single target site, is theoretically quite prone to the emergence of fungicide resistance with long-term use, and it is expected that resistance to prothioconazole might rapidly develop as it has done in other triazole fungicides, including tebuconazole, epoxiconazole, and propiconazole. However, previous research has shown that many plant pathogens appear to possess more than one CYP51 homologue, including *F. graminearum*, which is known to have three [34]. Furthermore, a detailed investigation found that the three homologues appeared to respond to DMI fungicides differently, as deletion of FgCYP51C appeared to have no effect on sensitivity at all, whilst deletion of FgCYP51A only increased sensitivity to tebuconazole, epoxiconazole, prochloraz, and propiconazole, and deletion of FgCYP51B increased sensitivity to all of the DMI fungicides tested [34], which indicates that triazole fungicides primarily target FgCYP51B, whilst other DMI fungicides target both the A and B homologues. It is perhaps not surprising, then, that mutations in the FgCYP51B amino acid sequence have frequently been associated with resistance to DMI fungicides. For example, Qian et al. found that the Y137H mutation in FgCYP51B was associated with tebuconazole resistance, and further that the tyrosine at residue 137 played a critical role in the tebuconazole binding pocket [25], while Zhao et al. discovered that the Y123H mutation reduced sensitivity to prochloraz by decreasing its binding affinity for FgCYP51B, and caused altered expression of the *FgCYP51B* gene [30]. It was therefore interesting to note that the current study found that all three of the prothioconazole-resistant laboratory mutants (3-aR, 4-aR, and SZ-1-3R) had mutations in their FgCYP51B sequences, although not the Y137 or Y123H substitutions that were previously documented (Table 2). The novel FgCYP51B mutations included P74T, F77L, G405S, Y37N, Q326R, and Y230F, the latter of which was perhaps the most interesting, as the mutant in which it occurred (4-aR) had no mutations in its FgCYP51A sequence, indicating a strong likelihood that this substitution was responsible for the observed prothioconazole resistance. Meanwhile, the Y37N and Q326R substitutions also occurred in a mutant (SZ-1-3) that had no additional mutations in its FgCYP51A sequence, although further investigation is required to discover which of these mutations are responsible for the resistance, or Indeed if both mutations are required simultaneously. In contrast, the P74T, F77L, and G405S substitutions occurred in a mutant (3-aR) that also had two additional mutations (L16F and S35P) in its FgCYP51A sequence, and, likewise, further investigation is required in order to clarify the contribution each might make to the resistant phenotype, although it was interesting to note that this mutant (3-aR) had dramatically reduced mycelial growth and sporulation compared to the mutants that only had changes in their FgCYP51B sequence (Figure 1 and Figure 2). In addition, it was also interesting to observe that two of the prothioconazole-resistant mutants (3-aR and 4-aR) also had amino acid substitutions in their FgCYP51C sequence, and although these are unlikely to contribute to the resistance mechanism given the observations by Fan et al. that this homologue did not appear to have 14α-demethylase activity, they might still account for the different biological characteristics of the three mutants, as FgCYP51C does appear to play a critical role in pathogenicity [34]. Indeed, the mutants (3-aR and 4-aR) with altered FgCYP51C sequences had significantly reduced pathogenicity compared to the one (SZ-3-1) that did not (Figure 3). Similar to the study by Zhou et al. [29], the current study also found that the expression of the *FgCYP51* genes was altered in the resistant mutants (Figure 4). However, the patterns of altered expression were extremely complex, involving both up- and downregulation, and it was difficult to ascertain whether the altered expression contributed to the resistance phenotype, or whether it was a consequence of changes to the intracellular processing of the mutated FgCYP51 proteins, as might be indicated by the dramatic upregulation of the mutated FgCYP51A (L16F and S35P) in 3-aR when prothioconazole was present (Figure 4C).

Cross-resistance is known to dramatically affect the efficiency of fungicide applications, and thus poses a major threat to consistent agricultural production. As such, information regarding the cross-resistance profile of the target pathogens is a key aspect of resistance risk assessments [33]. Although a previous study demonstrated that there was no cross-resistance between prothioconazole and carbendazim, tebuconazole, phenamacril, and pydiflumetofen in *F. graminearum* [22], a similar study on the closely related species *F. pseudograminearum* did find evidence of cross-resistance with other DMI fungicides, such as prochloraz, metconazole, tebuconazole and hexaconazole, but not with the non-DMI fungicides carbendazim, phenamacril, fludioxonil, and azoxystrobin [33]. It was therefore extremely encouraging that the current study found no evidence of cross-resistance between prothioconazole and any of the other fungicides tested, which included carbendazim, pyraclostrobin, and fluazinam, as well as the triazole tebuconazole and the imidazole DMI prochloraz (Table 1), which was similar to the results of Liu et al., but contrasted to those of Wei et al. [22,33]. Of course, this result may also be due to the low resistance multiple of the tested; the prothioconazole-resistant mutants, 3-aR, 4-aR, and SZ-1-3R had EC_50_ values of 10.10, 12.34, and 21.24 μg/mL, respectively (Table 1). These results therefore have practical implications for the management of FHB, since even if resistance does emerge when prothioconazole is used as a frontline systemic fungicide, curative control can still be obtained through the application of other fungicides in the later stages of crop growth. Moreover, the use of prothioconazole either in combination, or alternation with tebuconazole, prochloraz, carbendazim, pyraclostrobin, and fluazinam could help mitigate the risk of resistance emerging in the first place, and thereby allow for ongoing and effective control of FHB that ensures high-quality and high-yielding wheat production for many years to come.

## Figures and Tables

**Figure 1 jof-09-01001-f001:**
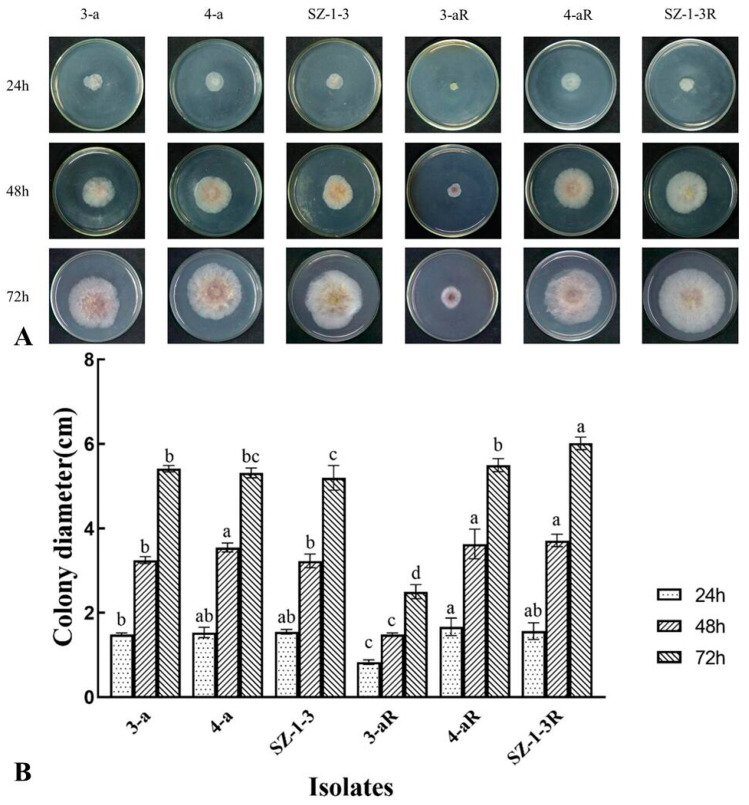
Mycelial growth of three prothioconazole-resistant mutants of *F. graminearum.* The top panel (**A**) shows the colony growth of both the resistant mutants (R) and their wild-type parental isolates on PDA after 24, 48, and 72 h incubation on PDA at 24 °C, while the graphs below (**B**) show the average colony diameter. Data represent the means of eight replicates ± standard error (SE). Different letters above columns indicate significant differences according to Fisher’s least-significant difference test (*p* ≤ 0.05).

**Figure 2 jof-09-01001-f002:**
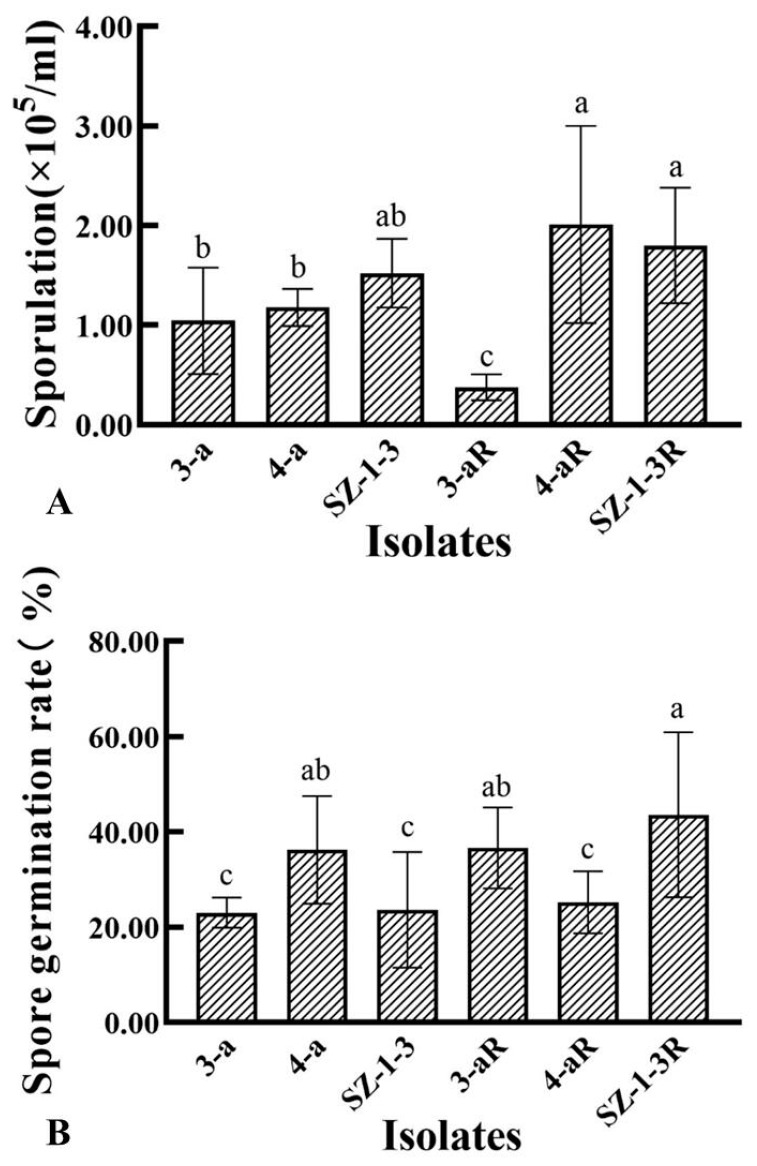
Sporulation and germination rates of three prothioconazole-resistant mutants of *F. graminearum.* The sporulation (**A**) of the three resistant mutants (R) and their wild-type parental isolates was assessed after incubation in mung bean broth at 24 °C with shaking (130 rpm) for 3 days, while the germination rate (**B**) was assessed on PDA after 24 h at 24 °C. Data represent the means of six replicates ± standard error (SE). Different letters above columns indicate significant differences according to Fisher’s least-significant difference test (*p* ≤ 0.05).

**Figure 3 jof-09-01001-f003:**
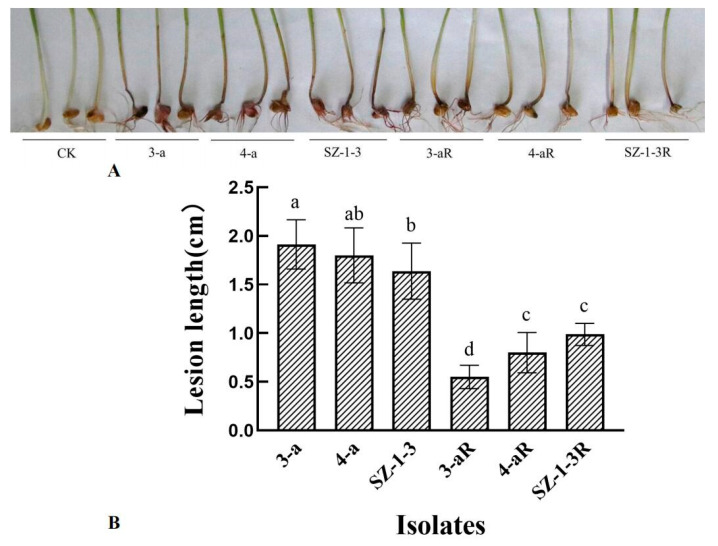
Disease symptoms and pathogenicity of three prothioconazole-resistant mutants of *F. graminearum*. Disease symptoms (**A**) caused by three prothioconazole-resistant mutants (R) were compared to those of their wild-type parental isolates at 14 days postinoculation (dpi), and the pathogenicity quantified based on the lesion length (**B**). Data represents the means of 10 wheat coleoptiles, and two independent experiments. Bars indicate one standard error (SE), while different letters above the columns indicate significant differences according to Fisher’s least-significant difference test (*p* ≤ 0.05).

**Figure 4 jof-09-01001-f004:**
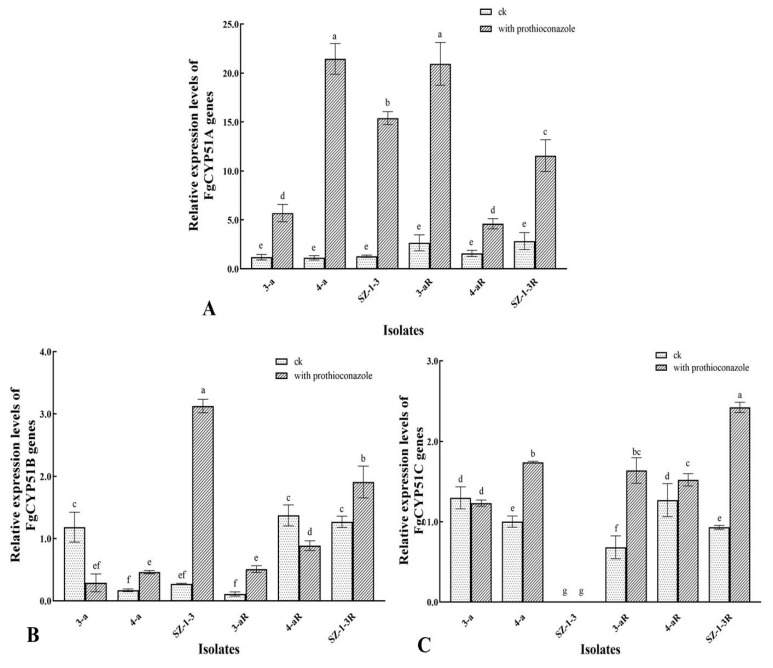
Relative expression of three *FpCYP51* genes in prothioconazole-resistant mutants of *F. graminearum.* The relative expression of three *FpCYP51* genes, including *FgCYP51A* (**A**), *FgCYP51B* (**B**), and *FgCYP51C* (**C**), was compared in both the resistant mutants (R) and their wild-type parental isolates, as well as in the absence and presence of prothioconazole (0.1 μg/mL), with β-tubulin as the reference gene. Different letters above the columns indicate significant differences according to Fisher’s least-significant difference test (*p* ≤ 0.05). Bars indicate one standard error (SE).

**Table 1 jof-09-01001-t001:** Cross-resistance between prothioconazole and five commonly used fungicides in three prothioconazole-resistant mutants of *F. graminearum*.

Fungicide	Sensitive Parental Isolates			Prothioconazole-Resistant Mutants		
	3-a	4-a	SZ-1-3	3-aR	4-aR	SZ-1-3R
Prothioconazole	0.58	0.31	0.25	10.09	12.34	21.24
Tebuconazole	0.02	0.04	0.03	0.04	0.04	0.02
Prochloraz	0.003	0.003	0.003	0.004	0.003	0.002
Carbendazim	0.24	0.21	0.20	0.26	0.26	0.25
Pyraclostrobin	0.47	0.13	0.27	0.26	0.38	0.38
Fluazinam	0.01	0.01	0.01	0.01	0.01	0.01

Values indicate the mean effective concentration (μg/mL) for 50% inhibition (EC_50_).

**Table 2 jof-09-01001-t002:** Amino acid substitutions occurring in the predicted sequences of three FgCYP51 subunits in prothioconazole-resistant mutants of *F. graminearum*.

Mutant	Nucleotide Changes	Silent Mutations	Amino Acid Changes	Gene
3-aR	T46C, C103T	A1496G	L16F, S35P	*FgCYP51A*
4-aR	/	T644C	/	
SZ-1-3R	/	/	/	
3-aR	C230A, T229C, G1381A	A54G, C753T	P74T, F77L, G405S	*FgCYP51B*
4-aR	T857A	C138T, G1182A	Y230F	
SZ-1-3R	T87C, A1144G	T522A	Y37N, Q326R	
3-aR	G545A, T627C, T821G, T873C, T1326A	T975, C1243T	E164K, I191T, S256A, M273L, V424E	*FgCYP51C*
4-aR	T975C	/	V307A	
SZ-1-3R	/	/	/	

/ Indicates no changes observed.

## Data Availability

Not applicable.

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
