# Peer review of "Exploring the Potential Mechanism of Prothioconazole Resistance in *Fusarium graminearum* in China"

_jof, 2023, doi:10.3390/jof9101001_

Round 1
Reviewer 1 Report
This is a well writing paper. I think it could be accepted with minor revision.
Minor points:
Line 14 should be deleted.
Whether prothioconazole is registered in China to control Fusarium graminearum or other wheat disease should be added in the Introduction.
Line 118-120: the concentration of solvents should be described.
Line 121: the collection information of these three isolates should be added.
The title of Figure 3B Disease strain length? Lesion length?
Line 247: silent mutations? Synonymous nucleotide mutation?
Line 241: the sequence results of these FgCYP51s could be added as a supplementary metairls.
Whether these point mutations found in this study had been reported in our DMI-resistant mutants except for Fusarium graminearum?
Why there is no cross-resistance between prothioconazole and other DMIs used in this study should be dicussced.
Minor editing of English language required.
Author Response
This is a well writing paper. I think it could be accepted with minor revision.
Minor points:
Line 14 should be deleted.
Response: Thanks for your careful review and helpful suggestions. We have revised the manuscript as suggested.
Whether prothioconazole is registered in China to control Fusarium graminearum or other wheat disease should be added in the Introduction.
Response: Thanks for your careful review and helpful suggestions. We have revised the manuscript as suggested.
Line 118-120: the concentration of solvents should be described.
Response: Thanks for your careful review and helpful suggestions. We have revised the manuscript as suggested, and the active ingredient content and manufacturers of the is added in the form of supplementary table 1. Meanwhile,the concentration of solvents also described in the manuscript.
Line 121: the collection information of these three isolates should be added.
Response: Thanks for your careful review and helpful suggestions. We have revised the manuscript as suggested.
The title of Figure 3B Disease strain length? Lesion length?
Response: Thanks for your careful review and helpful suggestions. We modify the title of Figure 3B as Lesion length.
Line 247: silent mutations? Synonymous nucleotide mutation?
Response: Thanks for your careful review and helpful suggestions. The amino acid mutations mentioned herein are synonymous nucleotide mutation.
Line 241: the sequence results of these FgCYP51s could be added as a supplementary metairls.
Response: Thanks for your careful review and helpful suggestions, and these results have been thoroughly analyzed and presented to readers in tabular form.
Whether these point mutations found in this study had been reported in our DMI-resistant mutants except for Fusarium graminearum?
Response: Thanks for your careful review and helpful suggestions. Most of the amino acid mutation sites are newly reported mutation sites that have not been previously reported in other plant pathogenic fungi.
Why there is no cross-resistance between prothioconazole and other DMIs used in this study should be discussed.
Response: Very good suggestion. As for the qualification problem, we have also thought deeply and analyzed it, which may be caused by the reason that the resistance ratio of the prothioconazole-resistant mutants we found was not particularly high (EC50 values of 10.10, 12.34 and 21.24 μg/mL).
Minor editing of English language required.
Response: Thanks for your careful review and helpful suggestions. We have revised the language of the manuscript as suggested.
Reviewer 2 Report
The authors have conducted a comparative study between Fg WTs and Prothioconazole mutants strain by physiological response, sequencing analysis, and gene expression. They found that two out of three mutants had positive growth and sporulation. Sequencing analysis (FgCYP51A-C) showed several nucleotide changes, but no conserved mutations were related to possible resistance. qPCR showed altered gene expression in WT and mutants by prothioconazole; no pattern correlation was observed. This report is premature research to consider a title related to the mechanism of Prothioconazole resistance in Fg. I recommended that the authors submit this research to another journal with less impact factor.
Change the title according to the results of the research.
L16-48 Consultee the instruction author to manuscript sections. The abstract is 200 words maximum.
L57: Which one region
L63: Delete “in China”
L73: delete “According to incomplete statistics reported in a 2020 publication, it was found that”.
L82: Delete “great”
There was no supplemental information available.
Author Response
Dear senior editor Ms. Su and reviewer 2:
Thank you for your letter and comments concerning our manuscript entitled “Mechanism of Prothioconazole Resistance in Fusarium graminearum in China” (ID: jof-2627892). These comments are all valuable and very helpful for revising and improving our manuscript. We have made comprehensive revisions of the manuscript according to you and the reviewer’s comments and suggestions. Revised portions have been marked in red in the manuscript. Meanwhile, the title has been updated to “Exploring the Potential Mechanism of Prothioconazole Resistance in Fusarium graminearum in China” according to reviewers’ suggestion.
Included are the point-to-point responses to reviewers’ comments.
Thank you very much for your hard work on our manuscript. Please don’t hesitate to contact us if you have some further questions.
Sincerely
Feng ZHOU
Reviewer 2:
Comments and Suggestions for Authors
The authors have conducted a comparative study between Fg WTs and Prothioconazole mutants strain by physiological response, sequencing analysis, and gene expression. They found that two out of three mutants had positive growth and sporulation. Sequencing analysis (FgCYP51A-C) showed several nucleotide changes, but no conserved mutations were related to possible resistance. qPCR showed altered gene expression in WT and mutants by prothioconazole; no pattern correlation was observed. This report is premature research to consider a title related to the mechanism of Prothioconazole resistance in Fg. I recommended that the authors submit this research to another journal with less impact factor.
Response: Thanks for your careful review and helpful suggestions. The current study indicated that three highly resistant laboratory mutants were exhibited significantly (p < 0.05) altered growth and sporulation, although contrary to expectation only one of the mutants exhibited reduce growth and sporulation, while the other two exhibited a significant (p < 0.05) increase. Despite this, pathogenicity tests conducted using wounded wheat coleoptiles revealed that all of the mutants exhibited significantly (p < 0.05) reduced pathogenicity, indicating a substantial cost to fitness. Sequence analysis of the prothioconazole target protein, CYP51, of which F. graminearum has three homologues (FgCYP51A, FgCYP51B, and FgCYP51C), identified three mutations in the FgCYP51B sequence with a high likelihood of being associated with the observed resistance, as well as another three mutations in the FgCYP51B sequence, and two in the FgCYP51A sequence that are worthy of further investigation. Two of the prothioconazole-resistant mutants were also found to have several amino acid substitutions in their FgCYP51C sequence, and it was interesting to note that these two mutants exhibited significantly (p < 0.05) reduced pathogenicity compared to the other mutant. Expression analysis revealed that prothioconazole treatment (0.1 μg/mL) resulted in altered expression of all the FgCYP51 target genes, and that expression was also altered in the prothioconazole-resistant mutants compared to their wild-type parental isolates. Meanwhile, no evidence was found of any cross-resistance between prothioconazole and other commonly used fungicides including carbendazim, pyraclostrobin, and fluazinam, as well as the triazole tebuconazole, and the imidazole DMI prochloraz. Taken together these results not only provide new insight into potential resistant mechanism in F. graminearum and the biological characteristics associated with them, but also convincing evidence that prothioconazole can offer effective control of FHB. We believe that this work will be of cross-disciplinary interest, because it not only indicated the biological characteristics of prothioconazole-resistant mutants, but also further indicated their molecular mechanism of prothioconazole resistance. As such, we think Journal of Fungi would be a suitable journal to bring our work to a wide and interested readership.
L16-48 Consultee the instruction author to manuscript sections. The abstract is 200 words maximum.
Response: Thanks for your careful review and helpful suggestions. We have revised the manuscript as suggested.
L57: Which one region
Response: Thanks for your careful review and helpful suggestions. We have revised the manuscript as suggested.
L63: Delete “in China”
Response: Thanks for your careful review and helpful suggestions. We have revised the manuscript as suggested.
L73: delete “According to incomplete statistics reported in a 2020 publication, it was found that”.
Response: Thanks for your careful review and helpful suggestions. We have revised the manuscript as suggested.
L82: Delete “great”
Response: Thanks for your careful review and helpful suggestions. We have revised the manuscript as suggested.
There was no supplemental information available.
Response: Thanks for your careful review and helpful suggestions. We have resupplemented the relevant attached documents and instructions.
Reviewer 3 Report
This study addresses Fusarium head blight (FHB) in wheat caused by Fusarium graminearum. Prothioconazole effectively controls F. graminearum. However, the emergence of prothioconazole-resistant strains poses a concern. The study generated three highly resistant laboratory mutants for investigation, observing altered growth and sporulation. Surprisingly, some mutants exhibited increased growth and sporulation. Despite this, all mutants displayed reduced pathogenicity, indicating a fitness cost. Genetic analysis of the target protein for prothioconazole, CYP51, identified specific mutations in FgCYP51B and FgCYP51A sequences, hinting at potential resistance mechanisms. Mutations in FgCYP51C, a non-14α-demethylation homolog critical for infection, were also noted in some mutants, correlating with reduced pathogenicity. Expression analysis highlighted altered FgCYP51 target gene expression in response to prothioconazole and in the resistant mutants. Notably, the study found no cross-resistance between prothioconazole and other common fungicides, suggesting its efficacy in controlling FHB, especially in combination with other fungicides. This research offers crucial insights into potential resistance mechanisms and biological characteristics of F. graminearum, emphasizing the importance of strategic fungicide use for ensuring successful wheat production.
However, several questions that remain to be answered or explored in future research
Why did you not carry out sampling in the fields treated with this fungicide on a large population in order to isolate resistant strains directly from wheat?
The abstract is very long and needs to be reduced
Line 114-120: Justify the choice of solvents
· explain the followed procedure to assess fungal growth and spore germination
· How do the studied mutations impact the mode of action of prothioconazole?
· Are there fungicides registered in China that could be explored for effective control of Fusarium wilt, particularly given the lack of cross-resistance with prothioconazole? What are the implications for crop protection strategies? What strategies can be implemented to minimize the emergence of resistance to prothioconazole and similar fungicides in F. graminearum? Are there any fungicides that combine prothioconazole with other active ingredients? Can such a combination be a viable strategy to mitigate resistance development while effectively managing FHB? How can such combinations be optimized for maximum efficacy?
· How does prothioconazole resistance affect the fitness and pathogenicity of F. graminearum?
· How can minimize or mitigate the potential environmental and health impacts associated with the extended use of prothioconazole and other fungicides in FHB management?
· You need to address some further research, such as identifying new targets for fungicides or exploring alternative, sustainable approaches to combat FHB effectively.
· Suggest integrated disease management approaches that combine fungicides with other agricultural practices, such as crop rotation, to provide a comprehensive strategy for controlling FHB
· Add a conclusion
Author Response
Dear senior editor Ms. Su and reviewer 3:
Thank you for your letter and comments concerning our manuscript entitled “Mechanism of Prothioconazole Resistance in Fusarium graminearum in China” (ID: jof-2627892). These comments are all valuable and very helpful for revising and improving our manuscript. We have made comprehensive revisions of the manuscript according to you and the reviewer’s comments and suggestions. Revised portions have been marked in red in the manuscript. Meanwhile, the title has been updated to “Exploring the Potential Mechanism of Prothioconazole Resistance in Fusarium graminearum in China” according to reviewers’ suggestion.
Included are the point-to-point responses to reviewers’ comments.
Thank you very much for your hard work on our manuscript. Please don’t hesitate to contact us if you have some further questions.
Sincerely
Feng ZHOU
Reviewer 3:
Comments and Suggestions for Authors
This study addresses Fusarium head blight (FHB) in wheat caused by Fusarium graminearum. Prothioconazole effectively controls F. graminearum. However, the emergence of prothioconazole-resistant strains poses a concern. The study generated three highly resistant laboratory mutants for investigation, observing altered growth and sporulation. Surprisingly, some mutants exhibited increased growth and sporulation. Despite this, all mutants displayed reduced pathogenicity, indicating a fitness cost. Genetic analysis of the target protein for prothioconazole, CYP51, identified specific mutations in FgCYP51B and FgCYP51A sequences, hinting at potential resistance mechanisms. Mutations in FgCYP51C, a non-14α-demethylation homolog critical for infection, were also noted in some mutants, correlating with reduced pathogenicity. Expression analysis highlighted altered FgCYP51 target gene expression in response to prothioconazole and in the resistant mutants. Notably, the study found no cross-resistance between prothioconazole and other common fungicides, suggesting its efficacy in controlling FHB, especially in combination with other fungicides. This research offers crucial insights into potential resistance mechanisms and biological characteristics of F. graminearum, emphasizing the importance of strategic fungicide use for ensuring successful wheat production.
However, several questions that remain to be answered or explored in future research
Why did you not carry out sampling in the fields treated with this fungicide on a large population in order to isolate resistant strains directly from wheat?
Response: Thanks for your careful review and helpful suggestions. We have down this work before, unfortunately, we did not isolate a field prothioconazole- resistant F. graminearum isolate. Indeed, further investigation revealed that the isolates were sensitive to prothioconazole with EC50 values ranging from 0.016-2.974 μg/m, and an average EC50 ± standard deviation (SD) of 1.191±0.72 μg/mL. And this work has been submitted to the journal of Plant Disease.
The abstract is very long and needs to be reduced.
Response: Thanks for your careful review and helpful suggestions. We have revised the manuscript as suggested.
Line 114-120: Justify the choice of solvents
Response: Thanks for your careful review and helpful suggestions. We have revised the manuscript as suggested, and the active ingredient content and manufacturers of the is added in the form of supplementary table 1. Meanwhile,the concentration of solvents also described in the manuscript.
Explain the followed procedure to assess fungal growth and spore germination.
Response: Thanks for your careful review and helpful suggestions. The procedure to assess fungal growth and spore germination as follows:
The mycelial growth of four sensitive wild-type F. graminearum isolates and their laboratory fludioxonil-resistant mutants was evaluated using a modified version of a protocol from a previous study (Zhou et al. 2023). Briefly, mycelial plugs (5 mm) were taken from the edge of 2-day-old colonies and transferred to fresh PDA plates that were then incubated at 24°C with a 12 h photoperiod. The resulting colonies were observed daily and the diameter of each measured at 24h, 48h and 72h post-inoculation. Each isolate was represented by six separate plates and the entire experiment performed once.
The rate of sporulation of the sensitive and resistant isolates was assessed as follows: the test colonies were initially established by transferring 5 mm mycelial plugs from 2-day-old PDA cultures to flasks containing 30 mL mung bean broth (MBB). After 3 days incubation at 24°C with shaking (130 rpm), the resulting spores were harvested and counted using a hemocytometer (Shanghai Qiujing Biochemical Reagent Instrument Co., Ltd). Each isolate was represented by at least three replicate flasks, and the entire experiment performed twice.
How do the studied mutations impact the mode of action of prothioconazole?
Response: Very good suggestion. Although FgCYP51B seems to play a key role as a sterol 14α-demethylase, a functional FgCYP51A gene could rescue mutants lacking FgCYP51B. Meanwhile, FgCYP51C is important in the synthesis of the deoxynivalenol (DON) toxin, and thus the pathogenicity of F. graminearum in wheat. In additional, further studies on whether these mutations of the FgCYP51A, FgCYP51B, or FgCYP51C are directly related to prothioconazole resistance through genetics and other techniques in F. graminearum are ongoing, and the next story will answer that question.
Are there fungicides registered in China that could be explored for effective control of Fusarium wilt, particularly given the lack of cross-resistance with prothioconazole? What are the implications for crop protection strategies? What strategies can be implemented to minimize the emergence of resistance to prothioconazole and similar fungicides in F. graminearum? Are there any fungicides that combine prothioconazole with other active ingredients? Can such a combination be a viable strategy to mitigate resistance development while effectively managing FHB? How can such combinations be optimized for maximum efficacy?
Response: Thanks for your careful review and helpful suggestions. In terms of the common understanding of this research field, cross resistance is known to dramatically affect the efficiency of fungicide applications, and thus poses a major threat to consistent agricultural production. As such, information regarding the cross-resistance profile of the target pathogens is a key aspect of resistance risk assessments. Although a previous study demonstrated that there was no cross-resistance between prothioconazole and carbendazim, tebuconazole, phenamacril, and pydiflumetofen in F. graminearum, a similar study in the closely related species F. pseudograminearum did find evidence of cross-resistance with other DMI fungicides such as prochloraz, metconazole, tebuconazole and hexaconazole, but not with the non-DMI fungicides carbendazim, phenamacril, fludioxonil, and azoxystrobin. It was therefore extremely encouraging that the current study found no evidence of cross-resistance between prothioconazole and any of the other fungicides tested, which included carbendazim, pyraclostrobin, and fluazinam, as well as the triazole tebuconazole and the imidazole DMI prochloraz. These results therefore have practical implications for the management of FHB, since even if resistance does emerge when prothioconazole is used as a frontline systemic fungicide, curative control can still be obtained by the application of other fungicides in the later stages of crop growth. Moreover, the use of prothioconazole either in combination, or alternation with tebuconazole, prochloraz, carbendazim, pyraclostrobin, and fluazinam could help mitigate the risk of resistance emerging in the first place, and thereby allow for ongoing and effective control of FHB that ensures high quality and high yielding wheat production for many years to come.
How does prothioconazole resistance affect the fitness and pathogenicity of F. graminearum?
Response: Thanks for your careful review and helpful suggestions. In the currently, two of the mutants (4-aR, and SZ-1-3R) had an increased growth rate on PDA, while the other (3-aR) displayed reduced growth. And a similar pattern was observed for sporulation, with 4-aR, and SZ-1-3R producing significantly (p < 0.05) more spores than their parental isolates, double the number in the case of 4-aR, while 3-aR produced significantly (p < 0.05) less, with a 70% reduction. However, it was found that the germination rate of the spores produced by 3-aR was actually significantly (p < 0.05) increased, whilst that of 4-aR was significantly reduced. Meanwhile, similar to 3-aR, the spores of SZ-1-3R also exhibited an increased rate of germination. Meanwhile, all of the prothioconazole-resistant mutants were found to exhibit a significant reduction in pathogenicity in comparison to their wild-type parental isolates, although the degree of change varied, with 3-aR exhibiting the most reduced lesions (≈70% smaller), and SZ-1-3R the least reduced (≈40% smaller). Taken together these results indicate that although prothioconazole resistance was associated with a certain cost to fitness.
How can minimize or mitigate the potential environmental and health impacts associated with the extended use of prothioconazole and other fungicides in FHB management?
Response: Thanks for your careful review and helpful suggestions. Scientific and rational use of prothioconazole to control Fusarium head blight. In particularly, be sure to pay attention to the correct dosage and frequency of prothioconazole application. Moreover, the use of prothioconazole either in combination, or alternation with tebuconazole, prochloraz, carbendazim, pyraclostrobin, and fluazinam could help mitigate the risk of resistance emerging in the first place, and thereby allow for ongoing and effective control of FHB that ensures high quality and high yielding wheat production for many years to come.
You need to address some further research, such as identifying new targets for fungicides or exploring alternative, sustainable approaches to combat FHB effectively.
Response: Very good suggestion. We will focus on this area of research in the next step.
Suggest integrated disease management approaches that combine fungicides with other agricultural practices, such as crop rotation, to provide a comprehensive strategy for controlling FHB.
Response: Very good suggestion. Effective control of Fusarium head blight is a systematic project, and chemical control is only one of the important measures. The prevention and control of Fusarium head blight involves the cross of many disciplines such as crop breeding, field cultivation management and chemical control. Of course, it is also the goal and direction of our further research in the later stage.
Round 2
Reviewer 2 Report
The authors have incorporated and improved the manuscript. It can be accepted in its present form.
"Exploring the Potential Mechanism of Prothioconazole Resistance in Fusarium graminearum in China "